# OpenReview forum: "Emotional Framing as a Control Channel: Effects of Prompt Valence on LLM Performance"
_ICLR.cc/2026/Conference — ICLR 2026 Conference Withdrawn Submission_

### Official Review · Reviewer_QGYZ · 2025-10-21

**Soundness:** 3
**Presentation:** 1
**Contribution:** 3
**Rating:** 2
**Confidence:** 4

**Summary:**

This paper investigates the influence of different emotional valences in prompts on LLM performance. The proposed evaluation framework comprises a MMLU-derived benchmark with diverse emotional tones and academic topics, and a structured rubric spanning multiple aspects. The authors claim that the well-aligned models are more robust against emotionally charged queries than misaligned models, suggesting that emotional robustness is a significant but underexplored metric in LLM alignment.

**Strengths:**

1. This paper proposes to study the emotional tone/framing of prompts, representing a valuable attempt to evaluate LLM performance in real-world settings. The three research questions presented in the introduction are well-designed and logically progressive.
2. A comprehensive evaluation suite is proposed, including a pipeline for benchmark data generation, a curated dataset, and a well-structured rubric.

**Weaknesses:**

1. Fig. 1 seems somewhat sketchy. It is recommended to revise this diagram figure to better distinguish the two pipelines, the there stages, and the different procedures.
2. Since there are more advanced sentiment classifier than vanilla BERT, why was BERT chosen for sentiment classification? Could the authors provide validation for BERT's sentiment judgments?
3. The abstract gives the impression that two datasets, i.e., the MMLU-derived one and a customized one, are used for evaluating LLMs, but according to Sec. 3.1, only the former is utilized.
4. I am skeptical about the classification of aligned and misaligned models. Considering that the training details of these LLMs are not transparent but inferred, for me, the most significant difference between the two groups may lie in model size rather than alignment extent.
5. The Tables 2-7 are difficult to read for me and occupy too much space. A more detailed analysis of key phenomena and generalized conclusions is needed so that readers from different backgrounds can better understand the implications of the reported statistics.

**Questions:**

1. Does ANOVA in L235 refer to "ANalysis Of VAlence"? Could you provide more insight into how to interpret results of the three analytical methods introduced in Sec. 3.4?
2. I recommend briefly introducing your evaluation protocal in Sec. 3.2, for example, clarifying what each rubric aspect measures, whether it relies on LLM-as-a-judge, and how potential bias was mitigated.

---

### Official Review · Reviewer_1ANU · 2025-10-27

**Soundness:** 2
**Presentation:** 2
**Contribution:** 2
**Rating:** 2
**Confidence:** 3

**Summary:**

This systematically studies how emotional tone—neutral, supportive, or threatening—affects large language model (LLM) behavior. Experimental results show that aligned LLMs remain stable, with valence influencing only stylistic features, while misaligned models are fragile. The study introduces emotional robustness as a missing dimension in current alignment frameworks and proposes prompt valence stress-testing as a diagnostic tool for assessing model safety and adversarial susceptibility.

**Strengths:**

1. The paper introduces a systematic framework and dataset for generating factually equivalent prompts with controlled emotional valence and graded intensity.

2. The experiments across six different LLMs are solid and sufficient.

**Weaknesses:**

1. I know it is not fair to reject a paper by writing quality, but the writing and organization of this draft need to be improved a lot.

2. The observation about aligned/unaligned LLMs is interesting, but superficial. I suggest authors conduct more experiments to answer why such a phenomenon exists? For example, is it because of the training data used in instruction tuning? or any tokens/concepts hidden behind the prompts?

3. No solutions are proposed to enhance the emotional robustness.

4. Related works for prompt sensitivity should be discussed in the draft.

**Questions:**

Please check the weaknesses.

---

### Official Review · Reviewer_Bhqi · 2025-10-27

**Soundness:** 2
**Presentation:** 2
**Contribution:** 2
**Rating:** 2
**Confidence:** 3

**Summary:**

The author show the effect of emotion/valence on the tone and factual content of model responses. They convert MMLU items into essay questions and score responses with LLM-as-a-judge. They find more stability across valence with "aligned" models, and in particular find degradation when misaligned models are prompted in "threatening" ways.

**Strengths:**

1. The writing and presentation are relatively clear.
2. There are some interesting possibilities raised by the aligned vs misaligned distinction.
3. The materials seem generally transparent and reproducible.

**Weaknesses:**

1. The paper seems to largely just show that valence affects model output, which seems like it is clear from common sense and everyday model use. The paper seems lacking in a deeper contribution, such as a new model of valence or the identification of unexpected effects of valence.
2. The distinction between "aligned" and "misaligned" models seems weak, missing a clear articulation of what alignment means (or why that term is used) and what systematic differences there are. As far as I can tell, there are too many idiosyncracies in these particular models to generalize like this.
3.It is unclear to me how exactly prompts were varied beyond (i) topic, (ii) valence, (iii) intensity. If there was variation, that should be more clearly documented. Showing how effects vary across those wording changes would also be important for a deeper contribution, such as showing more systematic effects of valence.

Other:
- The paper uses hyphens where it should use em dashes.

**Questions:**

1. How does this testing of valence effects advance our general understanding of machine learning?
2. Can we really clearly distinguish "aligned" and "misaligned" models? How are we isolating differences in alignment, and what exactly does that mean? Maybe this just needs clarification of wording to talk about base and post-trained models.
3. Why are there so many statistical analyses? It does not seem like we need "robustness cross-checks" here. I also don't see why we need non-parametric tests. These seem like large samples where we can just use straightforward linear models.

---

### Official Review · Reviewer_BhYE · 2025-11-01

**Soundness:** 2
**Presentation:** 2
**Contribution:** 2
**Rating:** 2
**Confidence:** 3

**Summary:**

This paper focuses on the emotional robustness of LLMs and conduct comprehensive analysis of how aligned and misaligned LLMs react to prompts with supportive, neutral and threatening prompts. For this purpose, the authors extract topics/queries from MMLU, crafted emotional templates, and utilize seven rubrics: relevance, factual accuracy, coherence, depth, linguistic quality, instruction sensitivity, and creativity, and construct 1,350 prompts. Using these prompts, this work analyzes several aligned and misaligned LLMs and concludes:  Aligned models are more stable across valences, while misaligned ones are not, with supportive prompts enriching style while the threats increase volatility.

**Strengths:**

1. This paper focuses on important and interesting research questions: How does prompt valence affect (aligned and misaligned) LLMs’ behaviors, and can emotional valence act as an adversarial channel.

2. This work conducts various analysis from an interdisciplinary perspective, and provided insightful conclusions, i.e., Aligned models are more stable across valences, while misaligned ones are not, with supportive prompts enriching style while the threats increase volatility.

**Weaknesses:**

The biggest problems lie in the experimental design.

1. The emotional prompts in Table 1 are not verified. How to ensure these prompts are associated with the ‘emotion’ for LLMs or to ensure LLMs’ behavior changes are caused by emotions in the prompts, instead of other confounders?

2. All the rubrics, e.g., relevance, factual accuracy, coherence, are evaluated using LLM-as-a-judge (line 154). However, the reliability of the LLM judges is not verified, which may influence all the derived conclusions.

3. The motivation becomes questionable upon the findings provided by this work. The authors claimed the proposed valence stress-testing can serve a diagnostic for alignment quality. It seems the conclusions show aligned LLMs are robust to emotional prompts, in line with our expectations. Since in most sensitive applications, most LLMs are well aligned, which limits the necessity and usability of this test.

4. There are a lot of missing references, for example:
    * Li et al., Large Language Models Understand and Can Be Enhanced by Emotional Stimuli. 2023.
    * Mozikov et al., EAI: Emotional Decision-Making of LLMs in Strategic Games and Ethical Dilemmas. 2024.
    * Huang et al., Emotionally Numb or Empathetic? Evaluating How LLMs Feel Using EmotionBench. 2023.

**Questions:**

1. In Line 34, the authors claimed “Users often communicate with artificial intelligence in emotionally charged ways-sometimes neutral, sometimes encouraging, sometimes frustrated or threatening” Are there any references/research to support such a claim? In my experience, users will not express emotions for AI as they know AI is only a tool.

2. What’s the reference paper [16]?

---

### Note · Authors · 2025-12-03

**Comment:**

We are withdrawing this submission. Thank you for your time and consideration.

**Withdrawal Confirmation:**

I have read and agree with the venue's withdrawal policy on behalf of myself and my co-authors.